# The Sensitivity of Structure to Ionic Radius and Reaction Stoichiometry: A Crystallographic Study of Metal Coordination and Hydrogen Bonding in Barbiturate Complexes of All Five Alkali Metals Li–Cs

**DOI:** 10.3390/molecules29071495

**Published:** 2024-03-27

**Authors:** William Clegg, Gary S. Nichol

**Affiliations:** 1Chemistry, School of Natural and Environmental Sciences, Newcastle University, Newcastle upon Tyne NE1 7RU, UK; g.s.nichol@ed.ac.uk; 2School of Chemistry, Joseph Black Building, University of Edinburgh, David Brewster Road, Edinburgh EH9 3FJ, UK

**Keywords:** crystal structure, alkali metals, barbituric acid, coordination, stoichiometry, bridging ligands, hydrogen bonding

## Abstract

A systematic study has been conducted on barbiturate complexes of all five alkali metals, Li–Cs, prepared from metal carbonates or hydroxides in an aqueous solution without other potential ligands present, varying the stoichiometric ratio of metal ion to barbituric acid (BAH). Eight polymeric coordination compounds (two each for Na, K, and Rb and one each for Li and Cs) have been characterised by single-crystal X-ray diffraction. All contain some combination of barbiturate anion BA^−^ (necessarily in a 1:1 ratio with the metal cation M^+^), barbituric acid, and water. All organic species and water molecules are coordinated to the metal centres via oxygen atoms as either terminal or bridging ligands. Coordination numbers range from 4 (for the Li complex) to 8 (for the Cs complex). Extensive hydrogen bonding plays a significant role in all the crystal structures, almost all of which include pairs of N–H···O hydrogen bonds linking BA^−^ and/or BAH components into ribbons extending in one dimension. Factors influencing the structure adopted by each compound include cation size and reaction stoichiometry as well as hydrogen bonding.

## 1. Introduction

The molecule barbituric acid (C_4_H_4_N_2_O_3_, Figure 1, henceforth BAH with singly deprotonated anion BA^−^) is the parent molecule of the class of drug compounds more generally known as ‘barbiturates’. It has no pharmacological activity but its 5,5-disubstituted derivatives do, particularly those with hydrocarbon substituents such as ethyl, butyl, phenyl, or cyclohexenyl groups. They have been extensively characterised by X-ray crystallography for many years and exhibit several interesting crystallographic properties, polymorphism and phase transitions being two of the major phenomena observed. Even if the search is restricted to unsolvated and simple solvates of derivatives with two identical 5-substituents, the Cambridge Structural Database (version 5.45, November 2023) [1] contains 11 structures including polymorphs [2,3,4,5,6,7,8,9,10,11,12]. The inclusion of cocrystals and of derivatives with two different substituents raises the number above 220. BAH is currently used in contemporary research as a model system for developing theoretical polymorph prediction techniques, something of major importance to the pharmaceutical industry [13,14,15,16].

The crystal structure of unsolvated barbituric acid was first determined in 1963 [17]; the study established that the six-membered ring is not perfectly planar, with the CH_2_ group displaced significantly from the mean plane of the other atoms, and this was attributed to crystal packing forces, particularly avoiding a short intermolecular C···O contact. A more recent redetermination of the structure confirmed this observation [15], and the report included a second polymorph containing two molecules in the asymmetric unit, one similarly slightly folded and the other essentially planar. A disordered high-temperature form (>233 °C) has been studied by powder diffraction [18].

The dihydrate of barbituric acid has been extensively studied. Room-temperature X-ray and neutron diffraction data indicate that planarity of the six-membered ring is required by all its atoms lying on a crystallographic mirror plane in space group *Pnma* [19,20,21]. We have found, through a series of 14 measurements of X-ray diffraction data over the temperature range 100–270 K, that the structure undergoes a phase transition on cooling at around 216–217 K to become monoclinic and twinned, with the removal of the crystallographic mirror plane (and other symmetry elements) and a significant out-of-plane displacement of the CH_2_ group and the two water molecules [22]. A more recent spectroscopic and theoretical study has suggested a subtle situation in which the room-temperature structure may be a thermally disordered state only approximating to the apparent higher orthorhombic symmetry and that there is no true phase transition [23]. The structure of this relatively simple molecule is thus more complex than first thought.

The limited flexibility of the barbituric acid molecule and its well-defined pattern of potential hydrogen bond donors and acceptors make it and its derivatives suitable candidates for crystal engineering applications. The extensive range of known cocrystal structures mentioned above, together with those of other barbiturates, includes, in particular, systematic studies of barbiturates with melamine derivatives [24,25,26,27].

Deprotonation of barbituric acid occurs at the CH_2_ group, as the resulting negative charge can be delocalised over the two adjacent carbonyl groups (Figure 1). BA^−^ is found as an uncoordinated discrete anion in over 40 reported crystal structures with organic, inorganic, and metal-centred cations; a full list of CSD REFCODES is given in the Appendix A. Hydrogen bonding plays an important role in these structures. BA^−^ is coordinated as a ligand to a transition or group 12 metal in only eight known crystal structures, some of which have been determined more than once [28,29,30,31,32,33,34,35,36]; there are no reported cases of coordination of BA^−^ to lanthanide, actinide, or *p*-block metals. Coordination is through oxygen in each case except for one copper complex, in which it is through nitrogen [29]. Published structures of BA^−^ complexes with alkaline earth (group 2) metals are restricted to only one of Ca [37] and one of Ba [38].

The number of previously published structures of complexes of BA^−^ with alkali metals (group 1) is also small. A set of one sodium and two potassium barbiturates, each also containing barbituric acid BAH in the crystal structure, has been reported, with crystal structures determined from room-temperature X-ray diffraction data [39]. A lithium complex has been structurally characterised from a twinned crystal at low temperature [40], while a rubidium barbiturate complex investigated at room temperature also contains BAH [41]. In view of the surprising scarcity of reported structures of metal complexes in which BA^−^ serves as a ligand and the lack of systematic studies in this area, we have carried out a detailed investigation of alkali metal barbiturate complexes for all five metals Li–Cs, prepared using only the metal carbonates or hydroxides and barbituric acid in boiling aqueous solution (except in one case), and varying the stoichiometry of the reagents to explore the possibility of obtaining different products. We report here a series of eight products (two each for Na, K, and Rb; one each for Li and Cs), all of which contain alkali metal ions coordinated by BA^−^ ligands. Four of them also contain BAH; five contain water, coordinated to the metal in every case while three are anhydrous. All eight have been characterised by single-crystal X-ray diffraction from data measured at 150 or 120 K and with consistent structure refinement strategies. Four of the structures are essentially the same as those reported previously, but almost all represent an improvement in precision. Conclusions are drawn regarding the influence of cation size and reaction stoichiometry on the structures of the products.

## 2. Results and Discussion

The reaction of barbituric acid with alkali metal carbonates and hydroxides in boiling aqueous solution, with some variation in reaction stoichiometry, has led in each case to products containing only the alkali metal cation, barbiturate anion, and, in some cases, barbituric acid and/or water. Charge balance requires a 1:1 ratio of M^+^ to BA^−^. None of the structures have uncoordinated species, and all coordination to the metal centres is through oxygen atoms. Some ligands are terminal, while others bridge two or more metal centres. A consistent pattern of atom numbering has been used for barbiturate and barbituric acid species in the series of structures, all of which are coordination polymers.

### 2.1. Compound ***1***, [Li(BA)(H_2_O)_2_]_∞_

Compound **1** crystallised from a 2:1 mixture of barbituric acid and Li_2_CO_3_, which corresponds to a 1:1 stoichiometric reaction ratio of BAH to Li^+^. In experiments to vary the reaction stoichiometry, the same product was obtained with half the quantity of BAH (1:2). The same compound was reported from a 1:1.6 ratio of BAH to Li^+^ (as the hydroxide) [40], so it appears to be the favoured product regardless of stoichiometry.

The coordination of Li is slightly distorted tetrahedral, with O–Li–O angles ranging from 101.39 (12) to 118.87 (13)°; details are given in Table 1 and the asymmetric unit, augmented to include the full coordination, is shown in Figure 1. The value of the angular distortion parameter τ_4_ is 0.92; it would be 1.00 for ideal tetrahedral geometry and 0 for square-planar [42]. Two water molecules serve as terminal aqua ligands, while two symmetry-equivalent BA^−^ anions act as bridges between pairs of cations, the ‘urea-like’ carbonyl group between the two N–H groups remaining uncoordinated. Bridging by the BA^−^ ligands generates coordination polymer chains in the [201] direction (not [101] as implicitly stated previously [40]).

The BA^−^ anion is planar with a root-mean-square (rms) deviation of <0.01 Å and has geometry typical of this species in previously reported structures, with a short C2–O2 bond of 1.2489 (17) Å and longer C1–O1 [1.2689 (17) Å] and C3–O3 [1.2630 (17) Å] bonds resulting from conjugation to support the charge of the anion.

Both N–H and all water O–H bonds serve as hydrogen bond donors; acceptors are all three O atoms of the anion and both water molecule O atoms (Table 2 of reference [40] erroneously lists no hydrogen bonding to one of the coordinated BA^−^ O atoms). Hydrogen bond geometry is given in Table 2. The hydrogen bonding connects the coordination polymer chains to generate a complex three-dimensional network. Essentially, parallel anions related by glide planes are stacked in the ***c***-axis direction with an interplanar separation of 3.477 Å, indicating a degree of π-π ring stacking. The stacking and hydrogen bonding can be seen in Figure 2.

### 2.2. Compound ***2***, [Na(BA)(H_2_O)]_∞_

In a synthetic procedure identical to that used for compound **1** but with sodium carbonate instead of lithium carbonate, and with the same variation of stoichiometry in different experiments (both 1:1 and 1:2 for for BAH:Na), compound **2** was obtained, having only one water molecule rather than two in the empirical formula. The simple formula belies a rather more complex crystal structure. The asymmetric unit contains one planar BA^−^ anion and one water molecule, but the required charge-balancing sodium ion consists instead of two Na^+^ ions, each lying on a crystallographic special position so that they effectively contribute half each to the asymmetric unit. One cation (Na2) lies on an inversion centre with a somewhat distorted octahedral geometry (*cis*-O–Na–O angles deviate up to almost 12° from the ideal 90°), while the other (Na1) lies on a twofold rotation axis and has a highly distorted six-coordinate geometry far removed from ideal octahedral. Each cation is coordinated by four different anions and two aqua ligands. The coordination geometry is given in Table 3 and the asymmetric unit, augmented to include the full coordination of both cations, is shown in Figure 3.

The BA^−^ anion coordinates through two of its three O atoms, but in this structure, the uncoordinated O atom is one of those adjacent to the C–H group (O3) and the urea-like carbonyl group O2 is coordinated to sodium; indeed, this atom is triply-bridging (μ_3_) to two symmetry-equivalent Na1 and one Na2 cations, while O1 coordinates only one cation, Na1. The aqua ligand (O4) forms a μ_2_ double bridge between Na1 and a symmetry-equivalent of Na2. The O2 and O4 bridges link sodium cations together into columns running along the ***c*** axis, and these columns are connected to each other by the anions, all of which are essentially parallel, as shown in Figure 4.

This three-dimensional coordination polymer is further supported by the π-π ring stacking of the anions in an arrangement very similar to that in compound **1** (interplanar separation 3.509 Å) and by hydrogen bonds involving both N–H and both aqua O–H bonds as donors and anion O1 and O3 as acceptors, as detailed in Table 4. The N–H···O hydrogen bonding generates ribbons of anions parallel to the ***b*** axis; such ribbons are typical of barbituric acid and related compounds such as cyanuric acid [39,43].

### 2.3. Compound ***3***, [Na(BA)(BAH)(H_2_O)_2_]_∞_

Compound **3** was obtained unexpectedly as one product in an attempt to prepare an iron complex of barbituric acid. The BAH:Na stoichiometry of the reagents was the same as in the synthesis of compound **2** (1:1), but the solution was more dilute and an aqueous solution of FeCl_2_ was added as a separate layer to give slow liquid diffusion. Iron is not incorporated into the product. The same compound has been reported as a product of a procedure apparently identical to our preparation of compound **2**, as well as from a sequence of kneading/grinding with BAH, Na_2_CO_3_, and water as starting materials in a range of stoichiometries [39]; the crystal structure was determined from room-temperature X-ray diffraction data, with a result of rather lower precision than the one we describe here.

The asymmetric unit contains one sodium cation, one BA^−^ anion, one BAH molecule, and two molecules of water as aqua ligands. Sodium has a distorted octahedral geometry, coordinated by the urea-like carbonyl O atom of BA^−^, the urea-like and one other carbonyl O atom of two symmetry-related BAH molecules, and three aqua ligands, two of them symmetry-equivalent. The anion acts as a terminal ligand, while BAH forms a bridge between pairs of cations. The aqua ligand O10 is terminal, while the pair of O9 aqua ligands bridge two cations to give a centrosymmetric, and hence planar, four-membered Na_2_O_2_ ring. The combined effects of these bridging and terminal ligands generate a coordination polymer chain in the [1$\overset{-}{1}$0] direction. The coordination geometry is given in Table 5, and a portion of the chain structure is shown in Figure 5.

The BA^−^ anion is planar, while the CH_2_ group of BAH is displaced out of the mean plane of the other atoms of this ring. The crystal structure does not feature any ring stacking. All N–H and O–H bonds serve as hydrogen bond donors, and the acceptors are all carbonyl O atoms except the urea-like carbonyl of BAH; no aqua O atoms accept hydrogen bonds. The hydrogen bonding links the coordination polymer chains to form sheets parallel to the ***bc*** plane (011), with no hydrogen bonding between sheets except a rather weak and non-linear O–H···O hydrogen bond between aqua O9 and the urea-like carbonyl O2. The N–H···O hydrogen bonds connect alternating BA^−^ anions and BAH molecules to form the familiar ribbons. Details of the hydrogen bonding geometry are given in Table 6, and a section of one hydrogen-bonded sheet is shown in Figure 6; here, the coordination polymer chains run diagonally from bottom left to top right, while the hydrogen-bonded BA^−^/BAH ribbons run vertically, with four overlapping pairs of them being visible in this Figure.

### 2.4. Compound ***4***, [K_2_(BA)_2_(BAH)(H_2_O)_3_]_∞_

The previously published research on alkali metal barbiturate salts and cocrystals [39] described three potassium compounds, of which two also contained barbituric acid and were characterised by single-crystal X-ray diffraction. They were obtained as two of three successive different crops of crystals resulting from a 1:1 stoichiometric ratio of K(BA) (prepared from the carbonate by grinding the carbonate and BAH in the presence of water) and BAH in aqueous solution, the first crop being a hydrate of BAH. The first of these two potassium compounds was [K(BA)(BAH)(H_2_O)_2_]_∞_, a formula analogous to that of the corresponding sodium compound (**3**) and with some similar structural features but a different overall crystal structure in detail; this compound has not been found in our investigations. The second potassium compound is the same as our compound **4**, prepared from a 1:1 stoichiometry for BAH:K in aqueous solution, and this has a lower content of both BAH and water (stoichiometry 1:1:0.5:1.5 instead of 1:1:1:2 for K:BA^−^:BAH:H_2_O) than in compound **3**. We have determined the crystal structure from low-temperature data and found that the asymmetric unit contains two potassium ions lying on a crystallographic mirror plane, one BA^−^ anion in a general position, one BAH molecule on a twofold rotation axis, and three aqua ligands with crystallographic mirror symmetry, corresponding to half of the chemical formula given in the section heading above. One of the two cations and one aqua ligand (coordinated to it) were disordered over two closely separated alternative positions with relative occupancy 0.6818:0.3182 (11), approximately 2:1. All the H atoms in this structure were refined freely apart from those belonging to the disordered aqua ligand, for which appropriate geometrical and displacement parameter constraints were applied. This disorder was not noted in the previously reported structure [39], which was derived from room-temperature data, but an inspection of the CSD-deposited results shows that it was an unresolved feature indicated by elongated displacement ellipsoids. The resolution of the disorder from low-temperature data permits a higher precision in the structural results. The discussion here is restricted to the major component of the disorder. It should be noted that we have chosen the unconventional space group setting *I*2/*m* in preference to the standard *C*2/*m*, as this gives a monoclinic β angle much closer to 90° for convenience in calculations and graphical displays. Once again, the anion is planar, and in this structure, the BAH molecule is also approximately planar as a consequence of lying on a twofold rotation axis.

The coordination geometry is given in Table 7. Figure 7 shows the asymmetric unit extensively augmented to include the full coordination of both potassium ions; minor disorder components are not shown. The two cations have different coordination arrangements: K1 is seven-coordinate with an irregular geometry, while K2 is eight-coordinate with an approximate square-antiprismatic geometry, the four symmetry-equivalents of O2 forming one square face and the other four coordinated atoms, the parallel square face rotated by 45°. K1 is coordinated by two BA^−^ anions, two BAH molecules, and three aqua ligands. K2 is coordinated by four BA^−^ anions, two BAH molecules, and two aqua ligands. The BA^−^ anion uses only one O atom for coordination; this is the urea-like carbonyl and it bridges three cations. By contrast, BAH bridges two cations through its urea-like carbonyl O atom, and coordinates to just one cation through each of the other two, which are equivalent by symmetry. Water is both terminal and bridging in this structure.

Columns of triply bridged potassium ions running parallel to the ***c*** axis are connected together by the BAH molecules, which are stacked with BA^−^ anions in the ***c***-axis direction, with a repeat sequence BAH··· BA^−^··· BA^−^··· and interplanar separations of 3.404, 2.370, and 3.404 Å. All the N–H and aqua O–H bonds serve as hydrogen bond donors, the acceptors being the uncoordinated O atoms of the anion and the singly-coordinated O atoms of BAH. The N–H···O hydrogen bonds generate the familiar ribbons connecting anions and BAH molecules. These, together with the columns of cations, can be seen in Figure 8, and the hydrogen bonding geometry is given in Table 8.

### 2.5. Compound ***5***, [K(BA)]_∞_

Changing the reaction stoichiometry to 2K:1BAH gives a different product with a much simpler empirical formula, a small unit cell, and an asymmetric unit containing only one cation and one anion, with no BAH and no water molecules. Although a compound with the same formula was previously obtained by grinding potassium carbonate and BAH in the presence of water [39], its crystal structure was not determined. It was reported as having only one half of the cation–anion pair in the asymmetric unit on the basis of the observed ^1^H, ^13^C, and ^15^N solid-state NMR spectra, so it does not seem to be the same material as our compound **5**, unless the NMR signals are fortuitously coincident peaks, an unlikely occurrence for the structure reported here.

Potassium has a distorted octahedral six-coordinate geometry. All three carbonyl O atoms of the anion are involved in coordination: the urea-like carbonyl group bridges three cations, one of the others bridges two cations, and the third coordinates only one cation. The coordination geometry is given in Table 9 and is shown in Figure 9. The structure is a three-dimensional coordination polymer.

The planar BA^−^ anions are π-π stacked in the ***a***-axis direction with an interplanar separation of 3.450 Å. There are only two crystallographically independent hydrogen bonds, the two N–H groups serving as donors to O1 and O3 and the urea-like O2 remaining uninvolved in hydrogen bonding. The hydrogen bonding, once again forming ribbons, is shown in Figure 10, with geometrical details in Table 10.

### 2.6. Compound ***6***, [Rb(BA)]_∞_

Compound **6** was prepared in essentially the same way as its potassium analogue, compound **5**, with a 2:1 stoichiometry of metal ion to BAH. Its empirical formula is the same, with Rb replacing K, but the crystal structure, while displaying some of the same features, has some important differences; this can be largely ascribed to the larger ionic radius of Rb^+^, which results in a higher coordination number of 7 instead of 6. The coordination geometry is irregular; details are given in Table 11. The augmented asymmetric unit is shown in Figure 11.

As in compound **5**, the urea-like carbonyl O atom bridges three cations, but here, both of the other two O atoms of the anion bond to two cations each, giving seven O–Rb bonds to match the coordination number of Rb. The cations and bridging O atoms generate tunnels with a parallelogram cross-section running along the ***b*** axis and these tunnels are connected by the planar anions, which are π-π stacked in the same direction with an interplanar separation of 3.437 Å, very similar to that in compound **5**. N–H···O hydrogen bonds, in which the acceptors are the doubly bridging O atoms, generate ribbons of the anions; geometrical details are given in Table 12. These features of the crystal packing can be seen in Figure 12. The similarity to compound **5** in Figure 10 is striking.

### 2.7. Compounds ***7*** and ***8***, [M(BA)(BAH)(H_2_O)]_∞_ with M = Rb, Cs

We describe these two compounds together, giving geometrical details for compound **8**, as they are isostructural. The rubidium complex **7** was obtained from a 1:1 stoichiometry of the cation and BAH, in contrast to the 2:1 ratio leading to compound **6**. A range of different reagent stoichiometries for caesium (as its hydrated hydroxide rather than the carbonate) led only to compound **8**.

A brief description of the structure of compound **7** has been published earlier [41], but the account of the synthesis from rubidium carbonate and BAH does not specify the quantities used. In that experiment, a mixture of restraints and constraints was applied to H atoms and the diffraction data were collected at room temperature, whereas we have used low-temperature data from a considerably smaller crystal and have refined all H atoms freely. Unlike the structures of compounds **1**, **3**, and **4**, our results here for **7** are not of higher precision.

The asymmetric unit contains one cation, one anion, one BAH molecule, and one aqua ligand. The coordination geometry for Cs is given in Table 13 and the augmented asymmetric unit is shown in Figure 13. The eight-coordinate arrangement has an irregular form that can be described as distorted pentagonal-bipyramidal with O2a and O2c in axial sites, an equatorial belt containing O1, O2b, O6e, and O9, and the fifth equatorial atom replaced by O5 and O5d displaced on opposite sides of the equatorial belt. Cs–O bond lengths cover the range 3.057 (4)–3.348 (8) Å. In compound **7**, the range of Rb–O bond lengths is even greater, 2.963 (2)–3.444 (2), the longest being to O2c in an axial site. Three of the eight coordination sites are occupied by symmetry-equivalents of O2, the urea-like carbonyl O atom of the BA^−^ anion, one by O1 of the anion, one by O6, the urea-like carbonyl O atom of BAH, two by symmetry-equivalents of O5 of BAH, and one by water serving as a terminal aqua ligand. Thus, O1 coordinates a single cation, O2 bridges three, O5 bridges two, and O6 coordinates a single cation. O3 in the anion and O7 in BAH are not involved in coordination.

The crystal packing of compound **8** is rather complex. Views down the intermediate ***c*** and short ***a*** axes are presented in Figure 14 and Figure 15. From these, it can be seen that oxygen-bridged Cs^+^ cations lie in columns parallel to the ***c*** axis and these columns are connected by the bridging BA^−^ and BAH ligands. The anions are planar, but the BAH molecules are twisted somewhat out of plane. There are no ring stacking interactions in the structure, the separations between pairs of six-membered rings all being above 4 Å. Hydrogen-bonded ribbons of alternating BA^−^ and BAH run parallel to the ***b*** axis and overlap each other when viewed down the ***c*** axis, so they are less clearly visible than in other structures of this series. These N–H···O hydrogen bonds involve all four N–H bonds as donors, the acceptors being O1 and O3 of the anion and O6 and O7 of BAH, but not O2 or O5, which bridge cations. The two aqua O–H bonds serve as donors to O3 and O7, each of which thus accept two hydrogen bonds. Hydrogen bonding geometry is given in Table 14.

### 2.8. Stoichiometry of Reactions and Products

Table 15 shows the stoichiometry of the reagents used and the stoichiometry found in the eight crystallographically characterised products in this investigation, together with the coordination number of the cations and the identity of the ligands contributing to the coordination in each case. A 1:1 ratio of cations and anions is required in all the products by charge balance; variation is found in the proportion of additional uncharged BAH and water in the structures.

At the two extreme ends of the series of alkali metals, only one product was obtained regardless of the stoichiometry of the reagents. In the case of Li, the incorporation of other species in addition to the cation and charge-balancing anion is limited by the small cation size and its consequent low coordination number, a tetrahedral geometry being common for Li^+^. Two coordination sites are occupied by bridging anions, and the coordination is completed by aqua ligands in preference to the bulkier BAH molecules. Cs^+^ has the largest ionic radius in the series and can readily accommodate a coordination number of eight, with BAH and water supplementing the four coordination sites occupied by multiply bridging anions.

For each of Na, K, and Rb, two different products have been obtained. In the case of sodium, these resulted from the same 1:1 ratio of Na^+^ to BAH and it is not clear what effect, if any, the FeCl_2_ has on the formation of compound **3**, a material previously obtained by other researchers as one of several products of a single experiment with a 1:1 stoichiometry of reagents [39]. Both **2** and **3** necessarily contain one anion for each cation, but **2** additionally has one aqua ligand, while **3** has one BAH and two aqua ligands. In both structures, sodium is six-coordinate, but in **2**, the coordination sites are occupied by four anions (which are multiply bridging) and two μ_2_ aqua ligands, while in **3**, they are occupied by only one anion, which is a terminal rather than a bridging ligand, together with two bridging BAH molecules and three water molecules as both terminal and bridging aqua ligands.

For each of K and Rb, two different reagent stoichiometries lead to two different products and there is a correlation between the ratio of reagents and the amount of BAH incorporated in the product. For both metals, a 1:1 ratio of M^+^ to BAH in the synthesis gives a structure including BAH and water, with a total of two neutral molecules per cation–anion pair, while a 2:1 excess of cation over BAH gives a simple M(BA) salt formula with no other chemical species present in the structure. Coordination numbers range from 6 to 8 for these cations, intermediate in size between Na^+^ and Cs^+^. Compounds **4** (K) and **7** (Rb) feature coordination of the cations by two or four anions, two or three BAH molecules, and one–three aqua ligands. The anions and BAH molecules all function as bridging ligands, but with varying numbers of O atoms engaging in coordination, while water is terminal in **7** but is both bridging and terminal in **4**. Regarding the different coordination numbers in these structures, it should be noted that one of the two crystallographically independent cations in compound **4** has five short (2.72–2.80 Å) and two long (3.00–3.03 Å) K–O bonds and the other has four short (2.67–2.72 Å) and four long (2.93–2.95 Å) K–O bonds, while the cation in compound **7** has six Rb–O bond lengths in the range 2.86–3.07 Å together with one of almost 3.28 Å and a particularly long one of 3.44 Å, which is longer than any of the eight Cs–O bond lengths in compound **8**. The high coordination in these two structures thus appears to be somewhat crowded. In compounds **5** and **6**, the absence of any BAH or water molecules necessarily means that the coordination numbers of 6 and 7, respectively, are satisfied entirely by multiply bridging anions. They have no notably long K–O or Rb–O bonds, the ranges being 2.64–2.81 and 2.81–3.17 Å, respectively.

### 2.9. Geometry of the Organic Ligands

All eight structures contain one BA^−^ anion in the asymmetric unit, in a general position with no imposed crystallographic symmetry. In every case, the anion is essentially planar, with root-mean-square deviations of the constituent atoms from the mean plane all under 0.04 Å. Four of the structures contain BAH molecules. In compound **4**, this lies on a crystallographic twofold rotation axis, which precludes displacement of the CH_2_ group out of the mean plane of the approximately planar ring. In compounds **3**, **7**, and **8**, the BAH molecule, in a general position with no imposed symmetry, has the CH_2_ group slightly displaced out of the mean plane of the other ring atoms.

The carbonyl groups of the anions and BAH molecules are involved to varying degrees in the coordination of cations in these structures, with the number of M–O bonds for individual carbonyl groups ranging from zero to three. It is of interest to compare the C–O bond lengths for different numbers of M–O bonds, between BA^−^ and BAH, and within individual ligands. Each organic species has three carbonyl groups, one of which is ‘urea-like’, lying between two N–H groups, and this is consistently labelled as C2–O2 in anions and C6–O6 in BAH molecules in all the structures; the other two are C1–O1 and C3–O3 in anions, and C5–O5 and C7–O7 in BAH molecules (C5–O5 and C7–O7 are symmetry-equivalent in compound **4**). The C–O bond lengths for all eight structures are collected in Table 16. For simplicity, standard uncertainties for the bond lengths are not shown; they lie in the range 0.0017–0.008 Å and full details are in the CIF files deposited at the CCDC. In the BAH molecules, all C–O bond lengths are essentially the same, with a total range of only 0.018 Å; there is no significant difference between the urea-like carbonyl and the others, and coordination to the cations has no discernible effect. Deprotonation of the BA^−^ anion slightly lengthens the urea-like carbonyl bond length and leads to a significant increase in the length of the other C–O bonds, as these are conjugated with the notional negative charge on the C–H group, as shown in Figure 1. As for the BAH molecule, coordination to the cations has no consistent impact on the carbonyl bond lengths.

## 3. Materials and Methods

All reagents were reagent-grade commercial products from Aldrich (Gillingham, UK) and Avocado (Altrincham, UK) and were used as received. Laboratory-grade distilled water was the only solvent for reactions. Elemental analyses were carried out by the Newcastle University Advanced Chemical and Materials Analysis unit. Access to suitable X-ray powder diffraction facilities was not available at the time of the experiments to confirm phase purity.

Synthesis of [Li(BA)(H_2_O)_2_]_∞_ (**1**): barbituric acid (0.264 g, 2 mmol) was dissolved in 15 cm^3^ distilled water. Solid Li_2_CO_3_ (0.074 g, 1 mmol) was added to the solution, which was heated to boiling. When the solid had dissolved and effervescence had ceased, the solution was set aside to cool. Slow solvent evaporation over a period of approximately two weeks resulted in large colourless lath-shaped crystals of **1** (0.066 g, 28.8%). Found C, 27.91; H, 3.97; N, 15.85%. C_4_H_7_LiN_2_O_5_ requires C, 28.25; H, 4.15; N, 16.47%.

Synthesis of [Na(BA)(H_2_O)]_∞_ (**2**): barbituric acid (0.254 g, 2 mmol) was dissolved in 15 cm^3^ distilled water. Solid Na_2_CO_3_ (0.105 g, 1 mmol) was added to the solution, which was heated to boiling. When the solid had dissolved and effervescence had ceased, the solution was set aside to cool undisturbed. After one month, colourless plate crystals of **2** were obtained (0.117 g, 69.6%). A reliable elemental analysis for the crystallographically characterised product could not be obtained due to loss of water when the crystals were stored in air. The results obtained correspond to the anhydrous barbiturate salt. Found C, 32.07; H, 2.01; N, 17.99%. C_4_H_3_N_2_NaO_3_ requires C, 32.01; H, 2.01; N, 18.67%.

Synthesis of [Na(BA)(BAH)(H_2_O)_2_]_∞_ (**3**): barbituric acid (0.260 g, 2 mmol) and solid Na_2_CO_3_ (0.110 g, 1 mmol) were dissolved in 30 cm^3^ distilled water. The mixture was boiled and stirred for 10 min and, while hot, was filtered into a hot conical flask. A solution of FeCl_2_ (0.129 g, 1 mmol) in 3 cm^3^ distilled water was introduced carefully down the side of the conical flask to allow slow diffusion of the two solutions and the flask was cooled slowly to room temperature. Crystals in two different habits were obtained after two days, small colourless blocks and very small brown needles. Owing to their small size, satisfactory crystallographic analysis of the colourless blocks was possible only with synchrotron radiation and showed them to be crystals of **3**. Mechanical separation of the crystal mixture for elemental analysis of the components was not possible and the brown needles were too small for single-crystal X-ray diffraction.

Synthesis of [K_2_(BA)_2_(BAH)(H_2_O)_3_]_∞_ (**4**): barbituric acid (0.260 g, 2 mmol) was dissolved in 15 cm^3^ distilled water. Solid K_2_CO_3_ (0.142 g, 1 mmol) was added to the solution, which was heated to boiling. When the solid had dissolved and effervescence had ceased, the solution was set aside to cool. After three days at room temperature, large colourless lath-shaped crystals of **4** were obtained (0.130 g, 25.5%). The elemental analysis results obtained correspond to a formula with loss of one water molecule after storage in air. Found C, 29.05; H, 2.75; N, 17.11%. C_12_H_14_K_2_N_6_O_11_ requires C, 29.03; H, 2.84; N, 16.93%.

Synthesis of [K(BA)]_∞_ (**5**): barbituric acid (0.126 g, 1 mmol) was dissolved in 15 cm^3^ distilled water. Solid K_2_CO_3_ (0.135 g, 1 mmol) was added to the solution, which was heated to boiling. When the solid had dissolved and effervescence had ceased, the solution was set aside to cool. Small clusters of colourless crystals of **5** grew in the sealed vial after one month (0.096 g, 57.8%). Found C, 28.86; H, 1.82; N, 16.54%. C_4_H_3_KN_2_O_3_ requires C, 28.91; H, 1.82; N, 16.86%.

Synthesis of [Rb(BA)]_∞_ (**6**): barbituric acid (0.131 g, 1 mmol) was dissolved in 20 cm^3^ distilled water. Rb_2_CO_3_ (0.232 g, 1 mmol) was dissolved in 10 cm^3^ cold distilled water and added to the barbiturate solution. After boiling until approximately 20 cm^3^ remained, the solution was set aside to cool undisturbed. Clusters of small colourless crystals of **6** grew over a period of one week (0.103 g, 48.5%). Found C, 22.67; H, 1.57; N, 13.11%. C_4_H_3_N_2_O_3_Rb requires C, 22.60; H, 1.42; N, 13.18%.

Synthesis of [Rb(BA)(BAH)(H_2_O)]_∞_ (**7**): barbituric acid (0.257 g, 2 mmol) was dissolved in 20 cm^3^ distilled water. Rb_2_CO_3_ (0.232 g, 1 mmol) was dissolved in 10 cm^3^ cold distilled water and added to the barbiturate solution. After boiling until approximately 20 cm^3^ remained, the solution was set aside to cool undisturbed. Slow evaporation of the solvent over two weeks resulted in clusters of small colourless crystals of **7** (0.158 g, 46.13%). Found C, 26.59; H, 2.54; N, 15.73%. C_8_H_9_N_4_O_7_Rb requires C, 26.79; H, 2.53; N, 15.62%. 

Synthesis of [Cs(BA)(BAH)(H_2_O)]_∞_ (**8**): barbituric acid (0.132 g, 1 mmol) was dissolved in 20 cm^3^ distilled water. CsOH·H_2_O (0.089 g, 0.5 mmol) was dissolved in 10 cm^3^ cold distilled water and added to the barbiturate solution. After boiling until approximately 20 cm^3^ remained, the solution was set aside to cool undisturbed. Slow evaporation of the solvent over four days resulted in clusters of small colourless crystals of **8** (0.103 g, 50.8%). Found C, 23.52; H, 2.21; N, 13.35%. C_8_H_9_CsN_4_O_7_ requires C, 23.66; H, 2.23; N, 13.80%.

X-ray crystallography: crystallographic experimental parameters for all eight compounds are summarised in Table 17 and Table 18. Data for compounds **2** and **4** were collected on a SMART 1K diffractometer (Bruker, Madison, WI, USA) using graphite-monochromated Mo-Kα radiation (λ = 0.71073 Å) at 150 K. Those for compound **3** were measured on an APEX2 diffractometer (Bruker, Madison, WI, USA) using synchrotron radiation (λ = 0.6814 Å) at 120 K. Otherwise, data were collected on a KappaCCD diffractometer (Nonius, Delft, The Netherlands) using graphite-monochromated Mo-Kα radiation at 150 K. Hydrogen atoms were refined freely with the following exceptions: for the H atoms of disordered aqua ligands in **4,** the isotropic displacement parameters were constrained to 1.2 times the equivalent isotropic parameters of the corresponding oxygen atoms, and the same constraint was applied to all H atoms of **6** and **8**; distance restraints were applied to the C-bonded H atoms of **6** and **8** [C–H = 0.95 (1) Å] and to the N- and O-bonded H atoms of **8** [N–H = 0.88 (1) Å, O–H = 0.84 (1) Å]. The crystals of **5** and **8** were found to be pseudo-merohedrally twinned. In the structure of **4**, one K^+^ cation and one aqua ligand were disordered over two sites each, with refined occupancy factors 0.682:0.318 (11); otherwise, the structures were fully ordered.

Software used: data collection with SMART [44], APEX2 [45], and COLLECT [46]; cell determination with SAINT [47] and DIRAX [48]; data reduction with SAINT and EvalCCD [49]; absorption correction by symmetry-equivalent and repeated reflections using SADABS [50]; structure solution by direct methods using SIR2002 [51] and SHELXTL [52]; refinement by full-matrix least-squares on all unique *F*^2^ values using SHELXL-2019/2 [53]; twinning analysis using ROTAX [54] and ROTWIN [55]. Structure figures were produced using SHELXTL and Mercury 2023.3.1 [56]. The structures have been deposited at the Cambridge Crystallographic Centre (CCDC), with deposition numbers given in Table 1 and Table 2; the data may be obtained at https://www.ccdc.cam.ac.uk/structures/ free of charge (accessed on 5 March 2024).

## 4. Conclusions

A number of trends can be seen and general comments made from this systematic survey of barbiturate complexes of the full set of alkali metals, Li to Cs. Although the eight structures described here have their own individual particularities, some features appear consistently. One of these is the presence of hydrogen-bonded ribbons of BA^−^ anions, also involving BAH molecules where these are present in the structure, based on pairs of N–H···O hydrogen bonds in an *R*^2^_2_(8) motif [57] linking adjacent rings. These are found in all the structures except that of compound **1**, which is the only one in which there are more aqua ligands than BA^−^/BAH. There are subtle differences in the form and composition of the ribbons. In compounds **3**, **7**, and **8**, anions and BAH molecules alternate along the ribbons, while in compound **4**, the sequence is BAH···BA^−^···BA^−^···BAH···BA^−^···BA^−^··· in accord with the stoichiometry of the structure. While the ribbons have the same overall shape in compounds **2**–**6**, shown at the top of Figure 16, a different shape is found in compounds **7** and **8**, shown at the bottom of Figure 16, whereby a urea-like carbonyl group (of BAH) is one of the acceptors, a feature not found in any of the other ribbons. In every structure, all the potential hydrogen bond donors are realised, while some acceptors remain unsatisfied except in compound **1**. Organic N–H groups and water O–H bonds form hydrogen bonds exclusively with carbonyl O atoms except in compound **1**, where water accepts both N–H···O and O–H···O hydrogen bonds. In terms of hydrogen-bonding patterns, then, compound **1** differs substantially from the other compounds in the series.

Two main factors can be identified as contributing significantly to the stoichiometry and structure of the compounds reported here. The first is the ionic radius of the cation, increasing down the group from Li to Cs; it clearly influences the coordination number, how many organic and aqua ligands can be accommodated around the metal centre. The second is the reaction stoichiometry, the relative numbers of organic and water molecules available to serve as ligands. The small size of the lithium cation and its typical tetrahedral coordination effectively rules out the incorporation of BAH molecules, even with a stoichiometric excess of this reagent. At the other end of the series, the preference for a high coordination of the large caesium cation makes the presence of BAH in the structure likely. For potassium and rubidium, different reaction stoichiometries lead to different products, BAH being included in the structure for a 1:1 ratio but not when the metal is in a 2:1 excess. The outcome of the syntheses proving most difficult to predict or rationalise is the amount of water found in the products, its absence or low proportion in some of the structures being compensated in coordination terms by enhanced bridging across two or three metal centres by the ligands present.

## Data Availability

Processed X-ray diffraction data are available as part of the information deposited at the Cambridge Crystallographic Data Centre. These may also be obtained from the author (W.C.) on request.

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
