# Peer review of "The Sensitivity of Structure to Ionic Radius and Reaction Stoichiometry: A Crystallographic Study of Metal Coordination and Hydrogen Bonding in Barbiturate Complexes of All Five Alkali Metals Li–Cs"

_molecules, 2024, doi:10.3390/molecules29071495_

Round 1

Reviewer 1 Report

Comments and Suggestions for Authors

The work by William Clegg, and Gary S. Nichol entitled “The sensitivity of structure to ionic radius and reaction stoichiometry. A crystallographic study of metal coordination and hydrogen bonding in barbiturate complexes of all five alkali metals Li–Cs.” systematically investigates the formation of barbiturate complexes involving all five alkali metals (Li–Cs) in aqueous solution, using variations in stoichiometry of metal ions to barbituric acid (BAH). Eight polymeric compounds, characterized by single-crystal X-ray diffraction, are identified. These compounds consist of combinations of barbiturate anion (BA−) in 1:1 ratio with the metal cation (M+), barbituric acid, and water molecules. Organic species and water molecules are coordinated to metal centers via oxygen atoms as terminal or bridging ligands. The structures exhibit coordination numbers ranging from 4 to 8, with extensive hydrogen bonding playing a significant role. The study elucidates factors influencing the structural diversity of the compounds, including cation size and reaction stoichiometry. BAH is currently used in contemporary research as a model system for developing theoretical polymorph prediction techniques, something of major importance to the pharmaceutical industry. The limited flexibility of the barbituric acid molecule and its well-defined pattern of potential hydrogen bond donors and acceptors make it and its derivatives suitable candidates for crystal engineering applications.

The work presents original data, although the concept of the work is very similar to a recent paper by authors published this year in Crystals (Ref. 55)

The research design combines state-of-the art methods and instruments.

The scientific importance is explained, but the range of applications could be explained in more detail.

The results are clearly presented, although the indications of bonds could distinguish covalent bonds.

The English language is fine and I detected no issues.

I have one remark that could be improved. In several places in the text, the authors call the solved structures complexed through alkali atoms and hydrogen bonds as “polymers” and “polymeric chains”. I believe, the word polymer is reserved to macromolecular chemistry, and denotes structures composed from units that are bound by covalent bonds. I suggest to drop from such use of terminology that may be confusing, given also the fact, the pictures use the same stick representation for covalent and non-covalent bonds.

After this minor issue is resolved, the manuscript can be published in the MDPI Molecules journal.

Author Response

This reviewer has one main recommendation regarding the designation of the compounds as polymers.  The term 'coordination polymers' has been extensively used since around 1950 to describe structures in which metal centres are connected by bridging ligands (organic or otherwise).  For a commentary on this usage, see K. Biradha, A. Ramanan and J. J. Vittal, Crystal Growth and Design 2009, 9, 2969-2970 (https://doi.org/10.1021/cg801381p).  This is the well recognised approach we have taken.  We have made minor changes in the text to ensure that our terminology is consistent throughout.  We have also, adopting the spirit of the reviewer's concerns, consistently depicted coordination bonds in the ellipsoid figures (one for each structure) in a way clearly distinguished from covalent bonds within the ligands.  Where additional dimensionality is generated in the structures by hydrogen bonding, we have avoided referring to this as part of the polymeric structure and have used the word 'network' to distinguish the types of interaction.  The modifications in the manuscript are highlighted in yellow (as for those in response to other reviewers' comments).

Reviewer 2 Report

Comments and Suggestions for Authors

Please see attached PDF document

Author Response

This reviewer makes 5 minor comments.

1. We did not use IR spectroscopy on these materials as our interest was essentially crystallographic.  The observation of vibrational frequencies would, we are sure, provide no significant additional information, merely supporting the well defined structural results.

2. We have added the suggested distortion parameter together with the relevant reference.

3. We have adopted the 'aqua ligand' terminology in general, though there are some places in the text where it is still appropriate to refer to water.

4. The synthesis procedures have been edited in accordance with this convention and the other points made by the reviewer.

5. We have applied italic formatting as suggested, and have superscripted the symmetry codes in the table bodies, but have retained the in-line (not superscripted) formatting of these letters in the table footnotes for greater clarity.  All ... symbols have been replaced by three centre-dots.

Reviewer 3 Report

Comments and Suggestions for Authors

In this work, the authors report all five alkali metals Li–Cs prepared from the metal carbonates or hydroxide in aqueous solution. And all compounds have been characterized by single-crystal X-ray diffraction. Thus, I do recommend its publishing after revising these following issues.

(1) The author needs to add the following content in the article: Why did the author do this work? What are the uses of the obtained compounds?

(2) Basic chemical properties need to be supplemented, such as TGA, IR, and stability in various solvents.

(3) The most important thing is that it is necessary to supplement some performance, as simple structural work is no longer meaningful.

(4) All tables except for 17 can be transformed into the supporting information.

Author Response

The points made by this reviewer are personal opinions rather than objective scientific requirements.  We have partially accepted the first, but do not accept the others.

1. The reason for the experimental work was already given in the Introduction section, but we have reworded this to make it clearer; the structural chemistry of complexes with barbiturate anions as ligands was remarkably sparse, offering an opportunity to provide significant and interesting new results.  The uses of barbituric acid derivatives are outlined in the Introduction; alkali metal 'salts' of simple organic acids such as this serve as synthetic intermediates in generating other complexes, but the research was not intended to address potential applications.

2. The report is a complete and self-contained structural study.  Investigating other aspects such as spectroscopic and thermal properties would be an optional extra dimension, but so would theoretical calculations and other possibilities.  Why should an investigation be all-encompassing?  This overlaps with the next point.

3. 'Simple structural work is no longer meaningful'.  We dispute this, while at the same time taking objection to the word 'simple' (echoes of the phrase sometimes heard 'only crystallography').  We agree that the report of a single crystal structure unrelated to anything else is, these days, not likely to be acceptable as a stand-alone publication - without a suitable context it has really nothing to say.  However, there are plenty of extended systematic structural studies of a whole series of related compounds that represent significant contributions to knowledge in structural chemistry and we believe (along, presumably, with the other reviewers) that this is such a case.  These structures are of intrinsic interest and the series as a whole displays clear trends.  They don't need to 'perform'.

4. This could perhaps be a valid point for a printed journal, to save paper, but it is not appropriate for a purely electronic publication, where the reader needs to look separately at the supporting information instead of having the information embedded in the main report.  All these tables are of selected material, the full geometrical results indeed being deposited.  The items selected for the embedded tables are described in the text and further illustrated in the Figures.